# Study on overburden failure law and surrounding rock deformation control technology of mining through fault

**Chunlin Zeng, Yuejin Zhou** **\*, Leiming Zhang, Donggui Mao, Kexin Bai**

State Key Laboratory for Geomechanics & Deep Underground Engineering, China University of Mining & Technology, Xuzhou, China

\* yuejinzh@163.com

## Abstract

In the mining process of working face, the additional stress generated by the fault changes the law of roadway deformation and failure as well as the law of overburden failure. Aiming at the influence of the fault in the mining process of working face, this study introduced the geological strength index (GSI) to analyze the stress distribution in the elastic-plastic zone of the surrounding rock of the roadway. And similar experiments under different engineering backgrounds were combined to study the characteristics of overburden movement and stress evolution. Based on the conclusions obtained, the roadway support scheme was designed. This study shows that, compared with ordinary mining, through-the-fault mining causes slippage and dislocation of the fault, the load of the overburden is transferred to both sides of the fault, and the stress near the fault accumulates abnormally. The "three zones" characteristics of the overburden movement disappear, the subsidence pattern is changed from "trapezoid" to "inverted triangle", and the influence distance of the advanced mining stress on the working face is extended from 20m to 30m. The instability range of roadway surrounding rock is exponentially correlated with the rupture degree of the surrounding rock. Through the introduction of GSI, the critical instability range of roadway surrounding rock is deduced to be 2.32m. According to the conclusion, the bolt length and roadway reinforced support length are redesigned. Engineering application shows that the deformation rate of the roadway within 60 days is controlled below 0.1˷0.5mm/d, the deformation amount is controlled within 150mm, and the roadway deformation is controlled, which generally meets the requirements of use. The research results provide guidance and reference for similar roadway support.

## 1. Introduction

Fault structure is a common geological structure in underground engineering, which affects mine safety production and underground engineering construction. The mechanical properties of the broken rock mass near the fault structure zone are obviously different from those of other rock masses. As the working face is advanced along the fault, the fault will interfere with

**Funding:** This research was supported by the Fundamental Research Funds for the Central Universities (2019ZDPY18 to JYZ) and the National Natural Science Foundation of China (51874289 to JYZ). The funders had no role in study design, data collection and analysis, decision to publish, or preparation of the manuscript.

**Competing interests:** The author(s) declare no competing financial interests.

the initial stress field, resulting in local subsidiary stress through the complexity and variability of the geological conditions near the fault [1]. In this case, fault activation is easily induced by mining stress. Due to the superposition of the initial stress and the mining stress, the stress environment becomes more complex, which leads to the unbalanced deformation of the roadway and even serious disasters [2–5]. Therefore, when the working face is advanced along the fault, a reasonable and effective support method should be adopted in the area affected by the fault to ensure the stability of the roadway and realize the safe and efficient operation of the mine.

Scholars at home and abroad have done a lot of research on the law of overburden failure and the stability of surrounding rock under the influence of faults. By studying the mining of fault footwall where steeply inclined ore bodies intersect, Atsushi Sainoki [6] found that the stress change caused by mining would lead to the fault activation. Qinglong Zhou [7] maintained that the stress concentration at the tip of the fault may cause the overburden movement, and quantitatively evaluated the stress concentration in the stress field at the tip of the fault. Qinglong Zhou [8] analyzed the failure trend of mining-induced fault based on mining-induced differential stress. Hongwei Wang et. al [9] simulated the fault movement in the process of coal mining by conducting similar experiments. On this basis, they concluded that the sharp increase of acoustic emission events could be used as the precursor of fault activation. Huiyong Yin [10] analyzed the movement law of coal seam roof and floor under different mining schemes by means of numerical simulation, which led to the conclusion that reasonable arrangement of mining sequence in the working face was an effective method to mitigate the influence of fault. Jeon [11] found that due to the existence of weak plane, the deformation of surrounding rock was asymmetric and the amount of deformation increased obviously. Donnelly [12] discovered that the stress distribution near the fault fracture zone played an important role in the safety and stability of mining operation, and the stress distribution of the long wall plate caused by mining exerted a significant impact upon the surrounding rock. In order to improve the integrity of the surrounding rock and the stability of the roadway, Liu Jun [13] and Fei Wan [14] et. al used the grouting reinforcement technology to study the surrounding rock control technology near the fault. By means of numerical simulation, Pu Wang [15] revealed the effects of strain energy release, fault slip and HTS failure settlement caused by obvious bending failure of HTS, which provided valuable reference for roadway support. Islam and Shinjo [16] studied the stress distribution and deformation of surrounding rock after the fault was activated by mining disturbance, and analyzed the stability and safety around the fault zone. These studies are based on the complex rock and soil environment under the influence of faults on the roadway failure. Due to the complex and changeable engineering geological conditions, the complex rock and soil environment and the impact of mining on the surrounding rock under the influence of faults are the main reasons for roadway failure [17, 18]. In order to ensure the stability of the surrounding rock of the roadway under the superimposed influence of faults and mining, it is necessary to study its failure mechanism, and then propose a reasonable support design.

The initial stress of fault itself and the influence of mining caused by underground engineering have great influence on the roadway and overburden [6, 19–21]. By means of the combination of theoretical analysis and similar experiments, this paper studies the characteristics of surrounding rock deformation and stress evolution law of overburden migration during the mining process under the influence of fault. On this basis, the arrangement and detection of supporting schemes was proposed, which further verified the accuracy of the conclusion obtained. This study provides valuable reference for the stability evaluation of surrounding rock and the design and construction of support structure during the process of coal mining.

## 2. The analysis of stress distribution characteristics around the roadway under the influence of fault

The fault zone is characterized by its broken state and the fracture development. Thus the nearby rock masses can be assumed as having the characteristics of jointed rock. For the large jointed rock mass which is not uniform due to joint fissures, Evert Hoek and E. T. Brown proposed and developed an empirical formula to predict rock fracture by introducing the geological strength index (*GSI*), which is shown as follows:

$$
\begin{cases}
\sigma_1 = \sigma_3 + \sigma_{ci}(m_b \dfrac{\sigma_3}{\sigma_{ci}} + s)^a \\
m_b = m_i e^{(GSI-100)/28} \\
s = e^{(GSI-100)/9} \\
a = \dfrac{1}{2} + \dfrac{1}{6}(e^{-GSI/15} - e^{-20/3})
\end{cases}
\tag{1}
$$

Where $\sigma_1$ and $\sigma_3$ denote the maximum and minimum principal stresses of the rock at the time of fracture, respectively; $\sigma_{ci}$ refers to the uniaxial compressive strength of rock; $m_b$ and $m_i$ denote the constant of rock fragmentation degree; a and s are the constant of material; *D* is rock disturbance coefficient.

A circular roadway can be taken as an example. The ring diameter of the critical range of instability is set as *R*, the radius of the roadway is set as l, and the distance between the center of the roadway and the elastic-plastic interface is set as *R*. Considering that the surrounding rock near the fault is in a stress equilibrium state under the combined action of radial stress ($\sigma_r$) tangential stress ($\sigma_\theta$) and shear stress ($\tau$), Hoek-Brown strength criterion is introduced as the criterion for judging the rock mass failure near the fault to analyze the instability range of surrounding rock [22, 23], as is shown in Fig 1.

### 2.1 Essential conditions

Assume that the original rock stress of the roadway is P, and the rock mass is isotropic and homogeneous, then the stress-strain relationship of the rock mass obeys Hoke's law in the

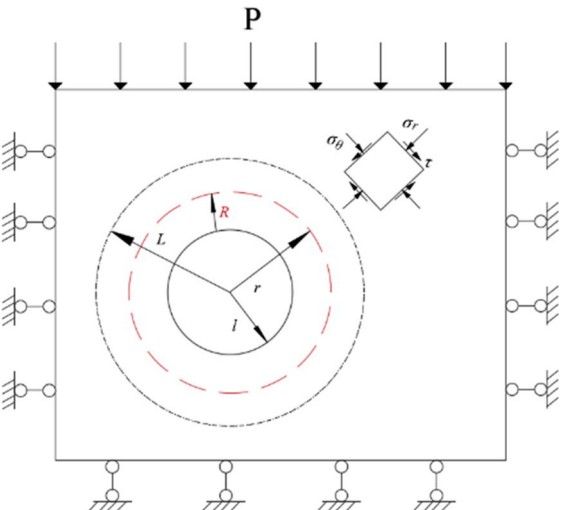

**Fig 1. Simplified mechanical model of roadway surrounding rock.**

elastic stage.

$$\begin{cases} \varepsilon_\theta = \dfrac{1}{E}[\sigma_\theta - \mu(\sigma_r + \sigma_z)] \\[2mm] \varepsilon_r = \dfrac{1}{E}[\sigma_r - \mu(\sigma_\theta + \sigma_z)] \end{cases} \tag{2}$$

Where $\varepsilon_\theta$ and $\varepsilon_r$ denote the tangential and radial strains in the elastic stage respectively; $\sigma_\theta$ and $\sigma_r$ denote the tangential and radial stresses in the elastic stage; $\sigma_z$ refers to the stress perpendicular to the roadway; $E$ stands for the elastic modulus of surrounding rock; $\mu$ is Poisson's ratio of surrounding rock.

When the surrounding rock yields, it satisfies the Mohr-Coulomb criterion. That is:

$$\sigma_\theta^p = K\sigma_r^p + \sigma_c^p \tag{3}$$

Where $\sigma_\theta^p$ and $\sigma_r^p$ denote the maximum and minimum principal stress of surrounding rock in the plastic zone; $\sigma_c^p$ refers to the plastics softening strength of surrounding rock.

$$K = \frac{1 + \sin \Theta}{1 - \sin \Theta} \tag{4}$$

where $\Theta$ denotes the internal friction angle of surrounding rock.

Under the condition of small deformation, the surrounding rock in plastic zone satisfies the stress differential equilibrium equation with gravity stress being ignored:

$$\frac{\mathrm{d}\sigma_r^p}{\mathrm{d}r} = \frac{\sigma_\theta^p - \sigma_r^p}{r} \tag{5}$$

## 2.2 The surrounding rock stress in elastic-plastic zones

**1) The surrounding rock stress in elastic zone.** According to Eq (3), it can be obtained:

$$\begin{cases} \sigma_r = \dfrac{E(\varepsilon_r - \varepsilon_\theta)}{1 + \mu} + \sigma_\theta \\[2mm] \dfrac{\mathrm{d}\varepsilon_\theta}{\mathrm{d}r} = \dfrac{1}{E}\left[\dfrac{\mathrm{d}\sigma_\theta}{\mathrm{d}r} - \mu\dfrac{\mathrm{d}\sigma_r}{\mathrm{d}r}\right] \end{cases} \tag{6}$$

Therefore, the surrounding rock stress in elastic zone can be shown as:

$$\begin{cases} \sigma_r = P + \dfrac{L^2(\sigma_L - P)}{r^2} \\[2mm] \sigma_\theta = P - \dfrac{L^2(\sigma_L - P)}{r^2} \end{cases} \tag{7}$$

**2) The surrounding rock stress in plastic zone.** By combining Eq (3) with Eq (5), the following equation can be obtained:

$$\sigma_c^P = r\frac{\mathrm{d}\sigma_r^P}{\mathrm{d}r} + (1 - K)\sigma_r^P \tag{8}$$

Let $M = r^{1-K}$, and then the above equation can be transformed into:

$$\frac{\mathrm{d}(M\sigma_r^P)}{\mathrm{d}r} = M\sigma_c^P r^{-1} \tag{9}$$

Accordingly, the surrounding rock stress in plastic zone can be shown as follows:

$$\sigma_r^P = \frac{1}{M} \int M\sigma_r^P r^{-1} \mathrm{d}r = r^{K-1} \int \sigma_r^P r^{-K} \mathrm{d}r \tag{10}$$

## 2.3 Analysis of critical instability range of roadway near the fault

According to the stress analysis of the surrounding rock, it can be known that the excavation of the roadway causes the redistribution of the stress field of the surrounding rock. Near the surrounding rock, the annular rock broken area is produced, that is, the instability zone in the roadway. In the instability zone, the relationship among the critical range of instability, the radius of the roadway, the distance between the center of the roadway and the elastic-plastic interface can be expressed by the following equation:

$$r = R + l \tag{11}$$

The excavation of the roadway changes the initial three-dimensional stress state of the surrounding rock from the elastic zone to the plastic zone. The surrounding rock mass is in a state of dynamic stress equilibrium, and its critical stress state is displayed as follows:

$$\frac{\mathrm{d}\sigma_r}{\mathrm{d}r} = \frac{\sigma_\theta - \sigma_r}{r} \tag{12}$$

The surrounding rock around the roadway is taken as the stress unit, and the radial stress and the tangential stress on it can be regarded as the maximum and minimum principal stress, respectively. By introducing the Hoek-Brown strength criterion, the following equation can be obtained:

$$\sigma_{ci} \ln r = \frac{\sigma_{ci}\left(\frac{m_b}{\sigma_{ci}}\sigma_r + s\right)^{1-a}}{m_b(1-a)} + C \tag{13}$$

At the elastic-plastic interface, the sum of the principal stresses in the elastic zone is equal to that of the plastic zone. Consequently, the boundary conditions can be obtained as r = l, $\sigma_r = 0$, $\alpha = 0.5$. Substitute these conditions into the above equation, and the following equation can be obtained:

$$C = \frac{2\sigma_{ci}\sqrt{s}}{m_b} - \sigma_{ci} \ln l \tag{14}$$

By combining Eqs (13) and (14) together, Eq (15) can be obtained:

$$\sqrt{\frac{m_b}{\sigma_{ci}}\sigma_r + s} = \frac{m_b \ln(r-l)}{2} + \sqrt{s} \tag{15}$$

let $N = \frac{m_b \ln(r-l)}{2} + \sqrt{s}$, and the critical limit stress of the plastic zone can be obtained:

$$\begin{cases} \sigma_r = \dfrac{\sigma_{ci}}{m_b}\left(N^2 - s\right) \\ \sigma_\theta = \sigma_r + \sigma_{ci}N \end{cases} \tag{16}$$

According to the critical limit stress state of the plastic zone, when the stress exceeds its limit value, the surrounding rock will be unstable. At the same time, at the interface between the elastic zone and the plastic zone, because the surrounding rock is in a dynamic equilibrium state, the critical limit stress of the surrounding rock in the plastic zone should be equal to that

in the elastic zone. In other words,

$$2P = \sigma_\theta + \sigma_r \tag{17}$$

By combining Eqs (16) and (17) together, the following equation can be obtained:

$$Pm_b = \sigma_{ci}\left[\left(\frac{m_b \ln(r-l)}{2} + \sqrt{s}\right)^2 - s\right] + \frac{1}{2}m_b\sigma_{ci}\left(\frac{m_b \ln(r-l)}{2} + \sqrt{s}\right) \tag{18}$$

Accordingly,

$$\frac{m_b \ln(r-l)}{2} + \sqrt{s} = \frac{-m_b\sigma_{ci} + \sqrt{m_b^2\sigma_{ci}^2 + 16\sigma_{ci}^2 s + 16 m_b\sigma_{ci}P}}{4\sigma_{ci}} \tag{19}$$

Thus, the critical instability range of roadway surrounding rock can be obtained, which provides a basis for the layout of engineering support scheme as follows:

$$\begin{cases} R = r - l = e^t \\ t = \dfrac{-m_b\sigma_{ci} + \sqrt{m_b^2\sigma_{ci}^2 + 16\sigma_{ci}^2 s + 16 m_b\sigma_{ci}P} - 4\sigma_{ci}\sqrt{s}}{2\sigma_{ci}m_b} \end{cases} \tag{20}$$

## 3. Simulation study of overburden movement under the influence of fault

### 3.1 Physical scaled model and parameter design

The 2401[#] working face across the S2F4[#] fault in No. 24 mining area of a coal mine in Henan Province was selected as the study background. In order to study the overburden failure law and the length range under the influence of the fault during the mining process of the working face, similar experiments were carried out by using the plane stress model. Since the roadway is in the same direction as the working face, so no roadway is arranged. The mining plane stress simulation device KD-01 developed by the State Key Laboratory of Deep Geomechanics and Deep Engineering, China University of Mining and Technology was employed; the size of the physical similarity model was 2.55 m×0.85 m×0.30 m; In total, there were 16 layers of strata which were divided into the left part and the right part by the fault; The fault dip angle was about 70°; The bottom and the top of the fault was about 0.70m and 1.00m away to the left boundary, respectively; The drop on both sides of the fault was 0.04m. Since the fault has a small drop and does not connect to the aquifer, the excavation of the working face under the influence of the fault is the main reason for roadway instability, so the groundwater is not considered. Fig 2 displays the schematic diagram of the model.

According to the formation lithology and the strength of similar materials, the experimental design scheme is shown in Table 1. The strata are horizontally arranged, with the geometric similarity ratio being 62.50, the bulk density similarity ratio being 1.47, and the stress similarity ratio is 91.88. According to the thickness of the covering layer is 428.11m and the stress similarity ratio is 91.88, a stress of 0.09 Mpa is applied to the upper surface of the model to compensate for the influence of the overburden load on the model.

### 3.2 Stress displacement monitoring and excavation design

As is shown in Fig 2, points at an interval of 5 cm are arranged on the surface of the model to serve as displacement measuring points. According to different mining methods, four stress monitoring points are set respectively. At the same time, since the periodic weighting length in the actual engineering cycle under ordinary mining conditions is 20 m, the distance between

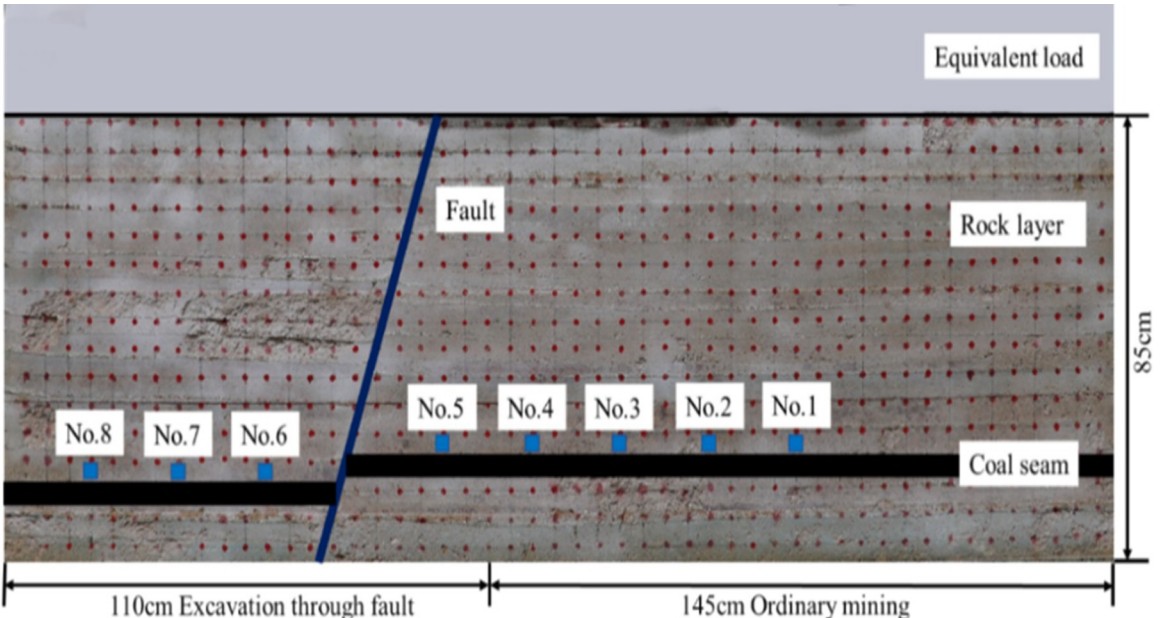

**Fig 2. Schematic diagram of stress measuring point layout.**

stress monitoring points is set to 20 cm. The model reserves 15 cm of boundary coal pillar to eliminate the mining boundary effect, and then advances from right to left with the mining step length of 5 cm. The normal excavation method is used before the model advances to 145 cm, and then shifts to the through-the-fault excavation method. The overburden movement and stress evolution in different mining scenarios are analyzed and compared to obtain the parameters of fault activation, rock caving characteristics, and stress concentration in the case

**Table 1. Experimental design of mixing ratio.**

| No. | lithology | Thickness of stratum/m | Thickness of model/cm | Material consumption/kg | | |
|---|---|---|---|---|---|---|
| | | | | River sand | calcium carbonate | gypsum |
| 16 | mudstone | 2.50 | 4 | 17.95 | 1.34 | 3.14 |
| 15 | medium sandstone | 3.81 | 6.1 | 61.10 | 7.33 | 17.11 |
| 14 | aluminous mudstone | 4.12 | 6.6 | 71.99 | 6.17 | 14.39 |
| 13 | siltstone | 4.18 | 6.7 | 70.47 | 7.04 | 16.44 |
| 12 | aluminous mudstone | 2.37 | 3.8 | 41.45 | 3.55 | 8.29 |
| 11 | siltstone | 1.87 | 3 | 31.55 | 3.15 | 7.36 |
| 10 | aluminous mudstone | 4.68 | 7.5 | 81.81 | 7.01 | 16.36 |
| 9 | medium sandstone | 1.75 | 2.8 | 28.05 | 3.36 | 7.85 |
| 8 | aluminous mudstone | 3.12 | 5 | 54.54 | 4.67 | 10.90 |
| 7 | siltstone | 4.75 | 7.6 | 79.94 | 7.99 | 18.65 |
| 6 | medium sandstone | 6.25 | 10 | 100.17 | 12.02 | 28.05 |
| 5 | siltstone | 3.60 | 5.7 | 59.95 | 5.99 | 13.98 |
| 4 | mudstone | 2.00 | 3.2 | 35.90 | 2.69 | 6.28 |
| 3 | coal | 2.40 | 3.8 | 43.60 | 2.90 | 6.78 |
| 2 | mudstone | 2.31 | 3.7 | 41.51 | 3.11 | 7.26 |
| 1 | fine sandstone | 6.00 | 9.5 | 88.14 | 8.81 | 20.56 |

of the through-the-fault excavation. On this basis, the roadway damage caused by through-the-fault excavation can be analyzed, which provides relevant experimental and data reference for the further studies on the deformation law and control of roadway surrounding rock.

## 4. Analysis of overburden movement law and stress evolution in different excavation scenarios

### 4.1 The overburden movement law in different excavation scenarios

As is shown in Fig 3, in the process of the working face advancing from the open-cut to the fault, when the working face is relatively far from the fault, the ground pressure characteristics of the working face are basically consistent with the law of overburden movement without the influence of faults, displaying obvious "three-zone" characteristics [24]. As is shown in Fig 3A, as the working face advances to 20 cm, the overhanging area of the direct roof increases continuously and the direct roof breaks for the first time. As is shown in Fig 3B, as the working face advances to 28 cm, the whole direct roof caves and the initial caving distance is 28 cm; as the working face advances to 40 cm, the separation appears between the direct roof and main roof; as the working face advances to 45 cm, longitudinal cracks appear at the edge of goaf. Fig 3C shows that the longitudinal cracks increase and develop gradually upward as the working face advances further; meanwhile, the roof bending and sagging becomes increasingly serious. Fig 3D shows that the first breaking of main roof occurs when the working face advances to 60 cm; the first roof weighting length of the main roof is 60cm. As is shown in Fig 3E, as the working face advances further, the periodic weighting length reduces to 30 cm. With the periodic weighting of the main roof, the height of caving zone above the coal seam is 145 cm, and the crack zone continues to expand upward. When the excavation reaches up to 70 cm, the longitudinal cracks at both sides of goaf and in the middle of goaf develop upward continuously; meanwhile a new separation appears about 55cm above the coal seam. The development height of the sagging zone has exceeded 50cm, and most of the load upon the overlying strata is transferred to the direction of the goaf and working face. Fig 3F shows that when the model is excavated to 85 cm, cracks occur in the aluminous mudstone layer, and longitudinal cracks develop further upward in the roof above the coal seam on both sides and in the middle of the goaf. At 76 m above the coal seam, new separation occurs under the siltstone layer, and the development height of the sagging zone has exceeded 70 cm. When the model is excavated to 100 cm, the cracks further develop upward to the upper plane of the model. At this time, the separation layer above the goaf is initially closed, and the overlying strata on the stope bend and sink synchronously with the advance of the working face, which is shown in Fig 3G. As is shown in Fig 3H, when the model is excavated to 110 cm, the separation layer above the goaf at the upper part of the stope is completely closed, and no new separation layer appears.

As is shown in Fig 4, when the working face advances to the influence range of fault, the overburden movement occurs violently, and the "three zones" characteristics of the overlying strata virtually disappear for all. Fig 4A shows that the top of the fault began to dislocate when the model is excavated to 145 cm; and the local activation height was 5cm. Fig 4B displays that the activation range of the local fault is enlarged, and some cracks on one side of the fault are closed when the model is excavated to 150 cm. Fig 4C shows that the activation range of local faults continues to increase when the model is excavated to 155 cm. Fig 4D displays that when the model is excavated to 160 cm, the working face mining enters the forceful through-the-fault stage. At this time, the fault is fully activated, with the footwall of the fault sinking and a wide gap appearing between the hanging wall and footwall of the fault. Ain the meanwhile, some cracks in the footwall of the fault are closed and no new cracks are generated. Fig 4E shows that when the model is excavated to 175 cm, the mining in the working face has been

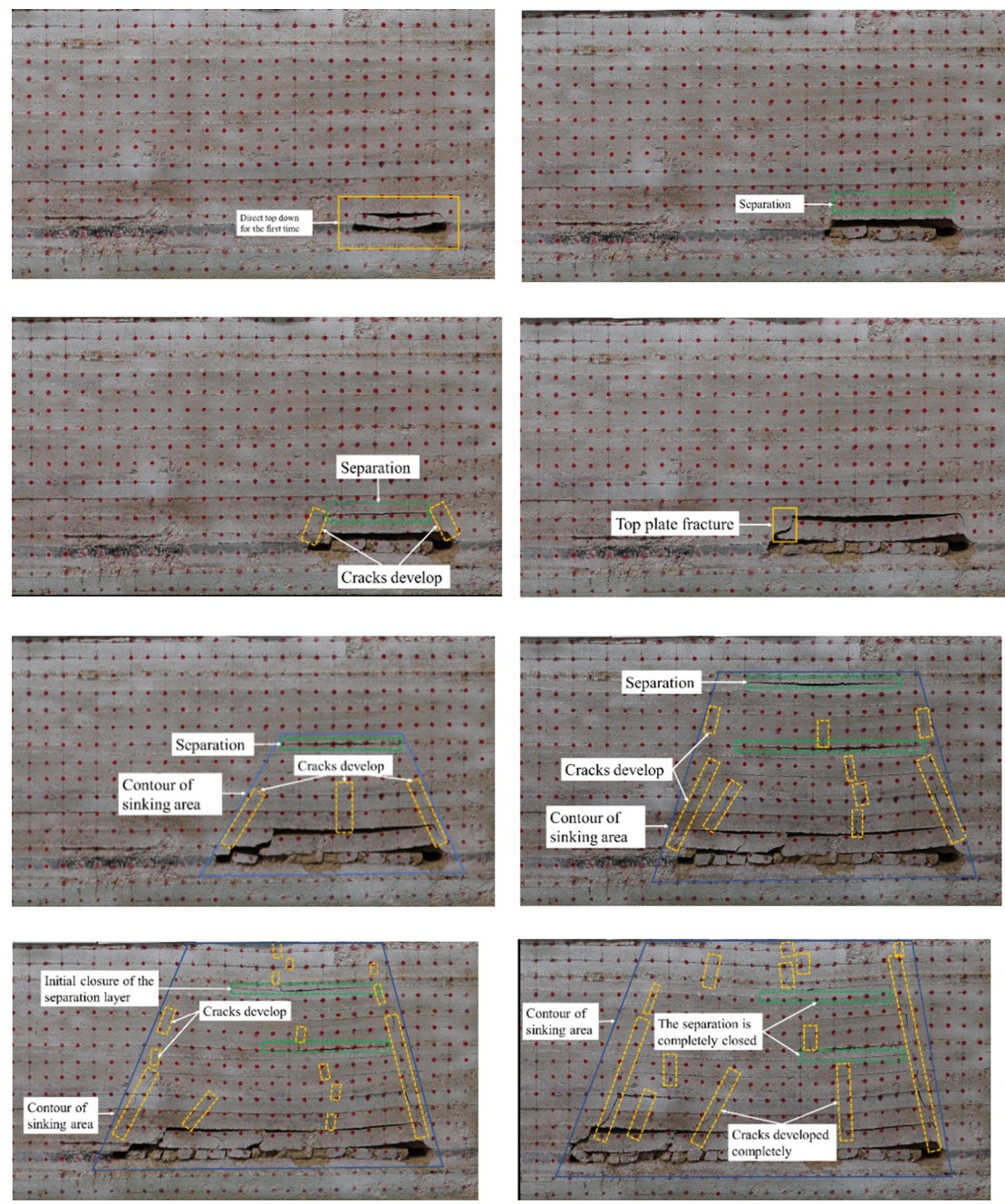

**Fig 3. Working face advancement diagram under normal mining conditions.** (a) The first caving of direct roof. (b) The appearance of separation. (c) The development of cracks. (d) The first breaking of main roof. (e) The cracks develop upward and form the separation. (f) The separation further develops upward. (g). The cracks develop up to the upper plane of the model. (h) The separation is completely closed.

out of the fault footwall range and enters the range of the hanging wall along the floor. The roof in the fault footwall caving zone sinks and fills the goaf. As is shown in Fig 4F, when the model is excavated to 180 cm, the roof breaks for the first time within the range of the fault hanging wall. When the model is excavated to 200 cm, the roof breaks for the second time within the range of the fault hanging wall, and the cracks begin to develop upward, as is shown

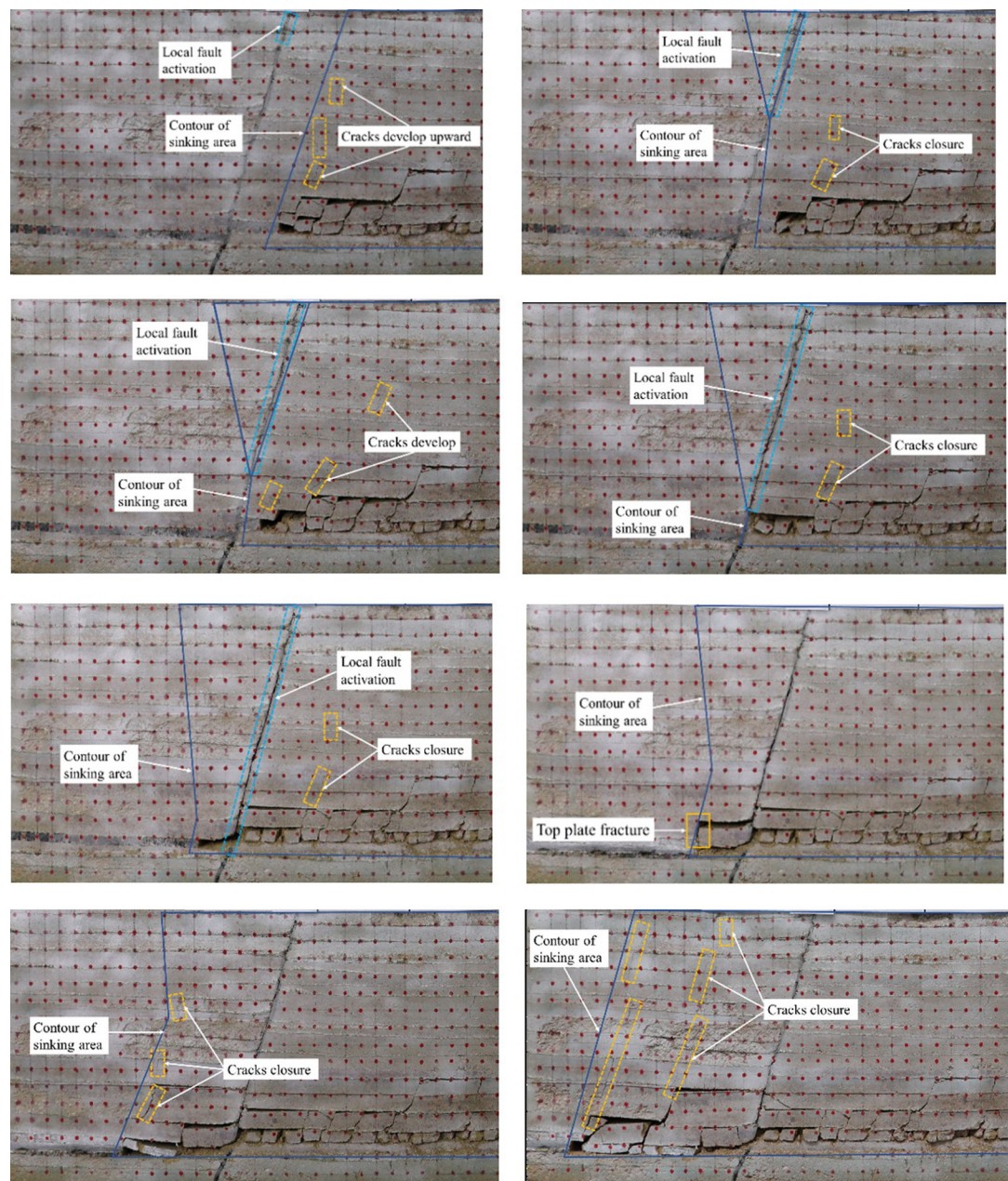

**Fig 4. Working face advancement diagram under normal mining conditions.** (a) Local fault activation occurs in the fault top. (b) Some cracks on one side of the fault are closed. (c) The activation range of the local fault continues to be enlarged. (d) The fault is fully activated. (e) The working face advances across the fault. (f) The roof breaks for the first time. (g) The roof breaks for the second time. (h) The model excavation is completed.

in Fig 4G. When the model is excavated to 225 cm, the roof breaks for the third time within the range of the fault hanging wall (as shown in Fig 4H). In other words, the periodic break distance of the roof on the fault hanging wall about 25~30 cm. Meanwhile, the cracks continue to develop upward and the overlying strata bend and sink as a whole. At this time, the model excavation is completed.

From the excavation of the model, the law of overburden movement under ordinary mining conditions can be revealed. To be specific, in the process of working face advancing, after

the initial collapse of the direct roof, the direct roof and the main roof in the caving zone above the coal seam continue to fracture and cave with the advance of the working face, presenting obvious periodical characteristics. Due to the continuous expansion of the goaf area, the overlying strata lose the support, and the separation is constantly generated in the overlying strata of the goaf area, and the formation is mostly located in the lower surface of the relatively thick and hard strata. With the advance of working face, the separation develops continuously until the goaf area is larger than the limit span of the thick hard rock; Then the thick hard rock starts to bend and cave in that the load on it exceeds its own bearing capacity, making the separation space gradually closed. There is a positive correlation between the development of overburden cracks and the advance of working face. The development of overburden cracks is in the form of inverted trapezoid from bottom to top, which ultimately transfers most of the overburden loads to the goaf. However, under the condition of mining through the fault, the characteristics of the three zones in the overlying rock virtually disappear. With the decrease of the distance from the fault, local activation occurs at the top of the fault first, and the hanging wall of the fault is broken in an inverted triangle, and the footwall is separated from the bed and compacted, and the crack is closed. When the excavation exceeds the footwall range of the fault, the overall activation of the fault occurs along the bottom floor of the strong over-fault, and the footwall completely sinks. Therefore, it is necessary to strengthen support in the fault range to prevent the roadway damage and guarantee the production safety.

## 4.2 The analysis of stress evolution in different excavation scenarios

According to the stress monitoring in each excavation sample section under normal mining conditions, the relationship curve between the stress value of each measuring point under normal mining conditions and the excavation distance is drawn, as shown in Fig 5A. With the advance of the working face, the stress in front of the working face increases and reaches the peak at the nearest place to the stress measuring point. When the working face exceeds the measuring point, the stress decreases rapidly to a certain value and then maintains a slight fluctuation. According to the above data, the stress generally begins to rise when the working face is about 30 cm away from the measuring point. In other words, the advance stress distance of the working face is about 20 m.

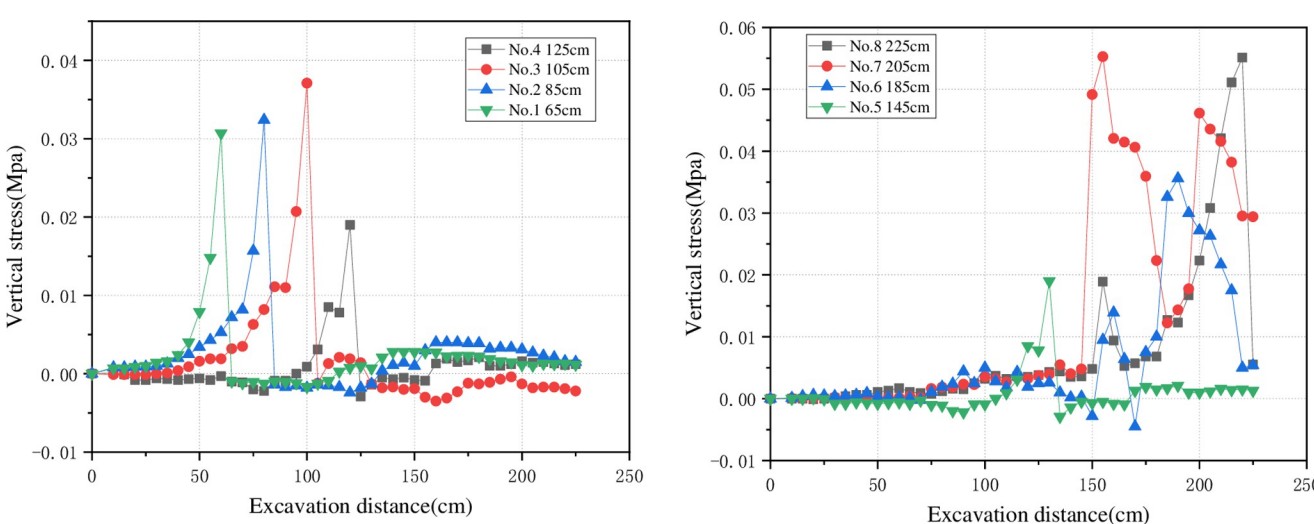

**Fig 5. Excavation distance-stress value curve.** (a) Normal excavation. (b) Through-the-fault excavation.

As is shown in Fig 5B, in the through-the-fault excavation scenario, in addition to the stress peak caused by the working face advancing near the measurement point, due to the activation of the fault, another stress peak appears in the vertical stress of the hanging wall of the fault when the excavation reaches up to about 155~160 cm. By comparing the overburden movement law in the stage of mining through the fault, it can be seen that the fault is fully activated when excavated to 155~160 cm, the hanging wall and foot wall of the fault are dislocated and separated, and the stress on both sides of the fault changes. During the advancement of the working face, the subsidence of the overlying rock layer changes from trapezoid to inverted triangle. A large amount of load migrates to the coal and rock mass of the hanging wall of the fault, causing the peak stresses at the measuring points of the hanging wall to be higher than that of the foot wall.

The vertical stress displays the following change trend: remaining stable in the pre-mining stage, rising during the mining stage, and declining in the post-mining stage. The mining distance is synchronized with the vertical stress change.

According to the relationship between the stress and the excavation distance, the abutment stress only fluctuates slightly when the working face is far from the stress measuring point, and there is no obvious stress change. When the model excavation advances to about 45 cm away from the fault, the vertical stress starts to rise sharply, corresponding to a 30 m-long influence range of advance support stress in actual engineering. When the model excavation approaches the fault, the stress reaches its peak value; and meanwhile, the influence of fault activation upon the roadway is very strong. When the working face exceeds the fault, the overlying strata collapse and the stress declines to a level slightly higher than the initial stress.

## 5. Engineering application

### 5.1 The general situation of the working face

The 2401[#] fully mechanized working face is located in the 24[th] mining area of a coal mine, with DF4 fault protecting coal pillar in the west, solid coal in the north, and F40[#] fault protecting coal pillar in the south. With a strike length of 184.9 m, the working face has a mining area of 147,095 m$^2$, and the average coal seam thickness is 2.4m. The working face is a monoclinal structure as a whole; the strata dips northwestward at a dip Angle of 5~15˚; and the average dip Angle is about 12˚. The whole working face strikes northeast to east northeast and dips northwest to north northwest. The direct roof is sandy mudstone with a thickness of 2.39~2.59 m, and the immediate floor is mudstone with a thickness of 0.5~1.2 m. During the excavation of the air return way and the haulage way, two normal faults are actually exposed. The distance between the two faults is 387.6m, and they do not affect each other during the mining of the working face, as shown in Fig 6.

### 5.2 The support design of roadway surrounding rock

According to the analysis of the actual geological conditions of the adjacent excavated roadways, the mining crossheading of the 2401[#] working face is greatly affected by the fault. As the working face advances towards the fault, the full activation of the faults will lead to abnormal increase of stress near the fault, which may seriously cause dynamic disasters such as rock burst. According to the results of the similar simulation experiment, with the mining face getting closer to the fault, the influence of the fault activity on the roadway increases significantly. Therefore, reinforced support is set within 30m from the fault. At the same time, according to the engineering geological conditions within this range, the relevant parameters of surrounding rock are set as follows: $GSI$—60, $l$—2.12 m, $\sigma_{ci}$—16 MPa, and $P$—9.0 MPa. Through theoretical analysis and by substituting these parameters into Eq (20), the critical instability range of roadway surrounding rock near the fault can be calculated as 2.32 m. According to similar

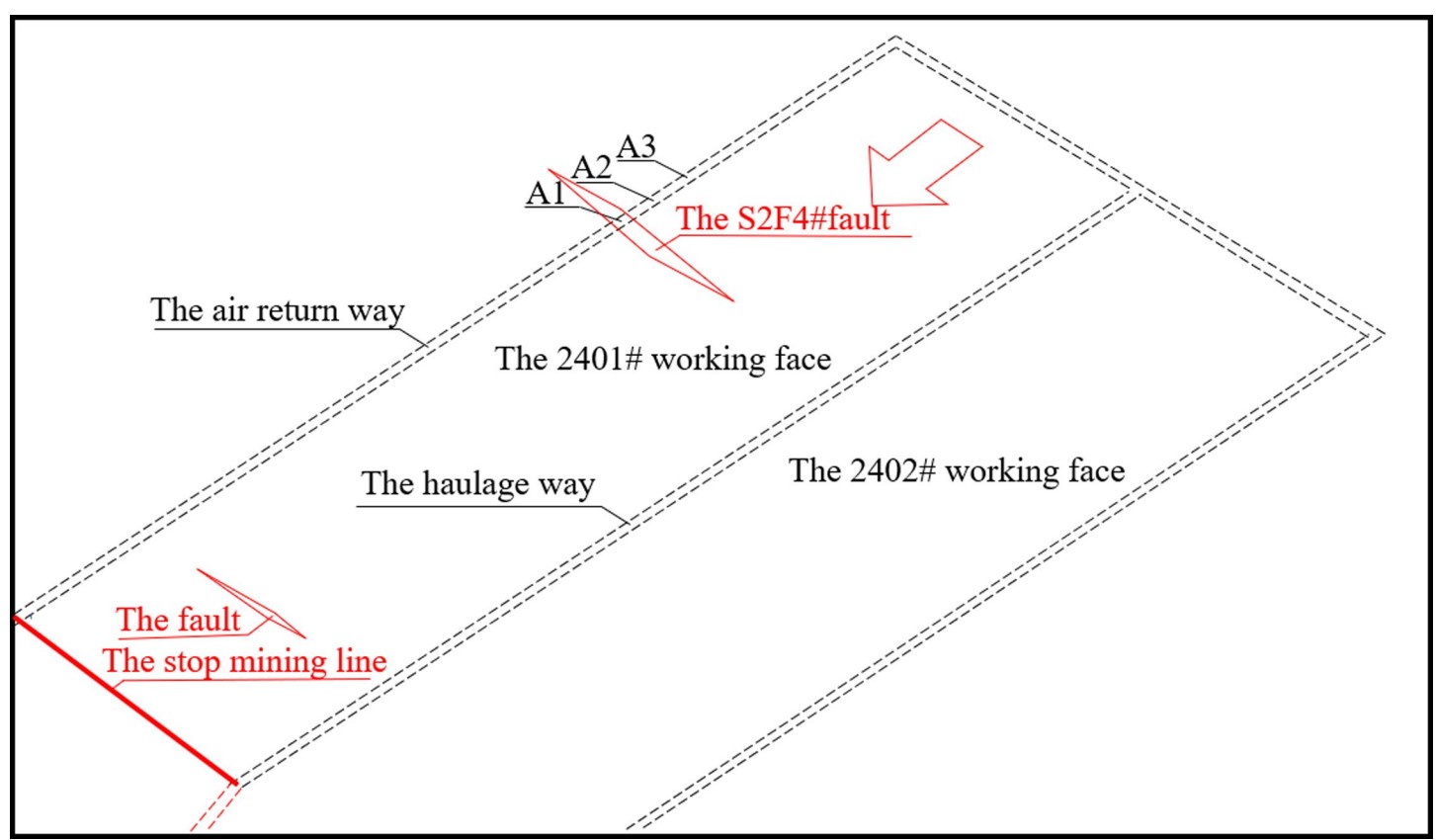

**Fig 6. The general situation of the working face and layout of measuring points.**

experiments and theories, the influence range of faults on the roadway in the mining process of the working face is obtained, and different supporting designs should be adopted for different mining scenarios.

**1) Normal mining support.** The support scheme under normal mining conditions is shown in Fig 7. To be specific, the diameter of the bolt is 22 mm, the length is 2.4m, the spacing between rows is 800 mm, the yield strength is 500 MPa, and the pre-tightening force is greater than 100 kN. The anchor cable adopts 4 strands of high strength and low relaxation steel strand with a diameter of 21.6 mm and a length of 7.3 m. Inter-row spacing is set be to 1,500 mm×1,400 mm, the pre-tightening force and anchoring force is greater than 100 kN and 200 kN, respectively. Metal mesh specification is 1,000 mm×2,000 mm, with the size of mesh being 100 mm×100 mm; Three anchor cables and 6 anchor rods are arranged on the roof, and the steel ladder made of steel rods with a diameter of 14mm is also hung on the roof.

**2) The support scheme adjacent to the fault.** The support scheme adjacent to the fault is shown in Fig 8. To be specific, the diameter of the bolt is 22 mm, the length is 2.6 m, the spacing between rows is 600 mm, the yield strength is 500 MPa, and the pre-tightening force is greater than 100 kN. The anchor cable adopts 4 strands of high strength and low relaxation steel strand with a diameter of 21.6 mm and a length of 7.3 m. Inter-row spacing is set be to 1,200 mm×1,400 mm, the pre-tightening force and anchoring force is greater than 100kN and 200kN, respectively. Metal mesh specification is 1,000 mm×2,000 mm, with the size of mesh being 100 mm×100 mm; Four anchor cables and 7 anchor rods are arranged on the roof, and the steel ladder made of steel rods with a diameter of 14 mm is also hung on the roof.

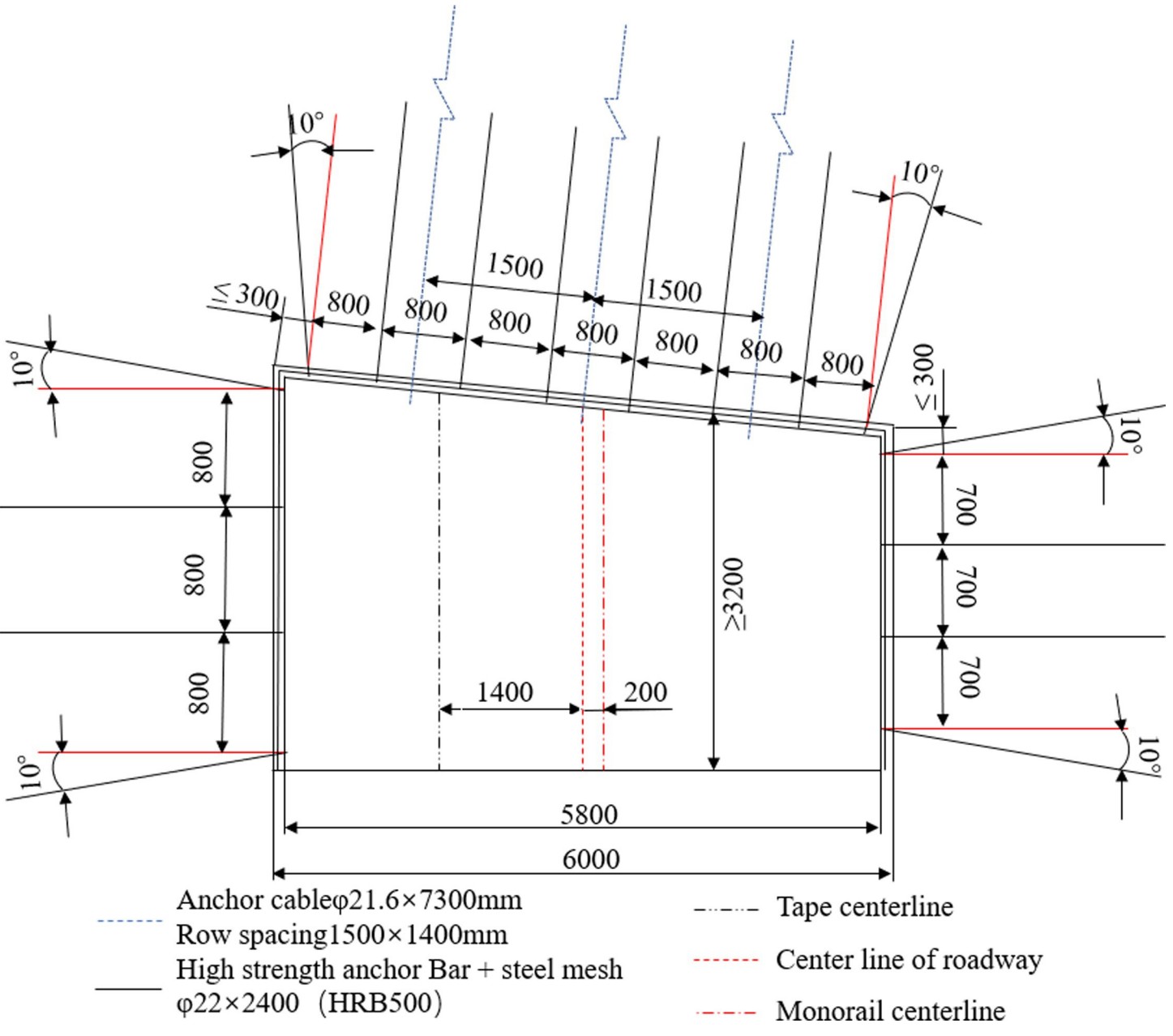

**Fig 7. Schematic diagram of roadway support.**

## 5.3 Analysis of roadway support effect

When the surrounding rock of the roadway is under the action of stress, it will experience multi-directional dynamic deformation. Therefore, the roadway stability can be intuitively judged by measuring the variation of floor heave horizontal convergence, and vault subsidence. For the air return way on the 2401[#] working face, measuring stations are set every 30 m starting from the fault position, and three measuring stations (A1, A2, A3) are set in total, as Fig 6 show. The whole observation on the measuring stations lasts for three months. The monitoring results are shown in Fig 9.

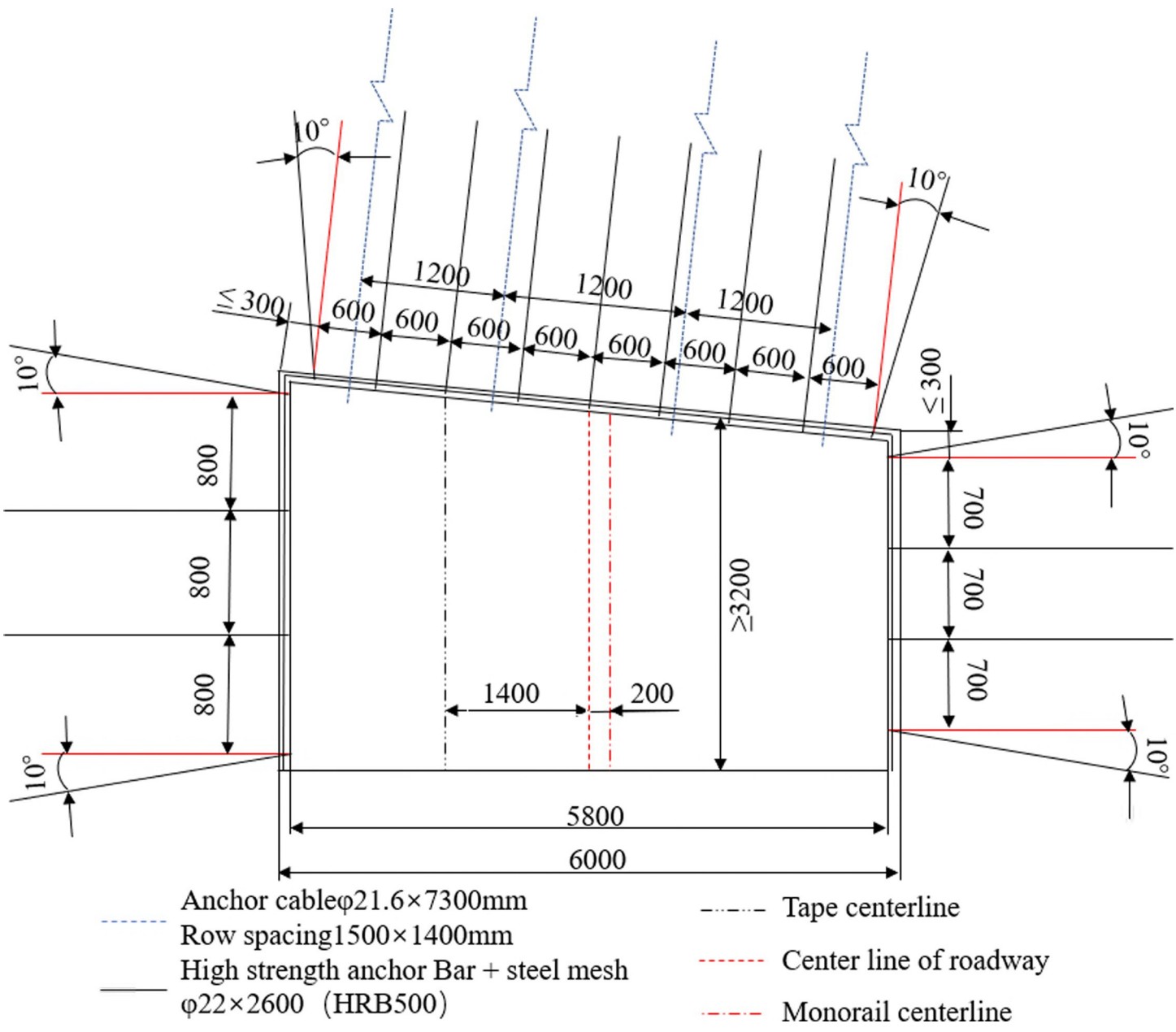

**Fig 8. The schematic diagram of roadway support adjacent to the fault.**

As Fig 9 show, the deformation rates of surrounding rock in the three roadway sections are below 0.2 mm/d; the variation trend of floor heave, horizontal convergence and vault subsidence observed at all the three measuring stations is basically consistent; the displacement variation rate increases rapidly at first and then decreases gradually, and finally the surrounding rock of the roadway tends to be stable. During the whole process, the stress of roadway surrounding rock experiences a constant and gradual adjustment. As the monitoring results obtained at A1 measuring station show, the average rate of horizontal convergence at two sides is 1.48mm/d, that of vault subsidence is 1.05 mm/d, and that of floor heave is 2.38 mm/d. The monitoring results obtained at A2 measuring station show that the average rate of horizontal

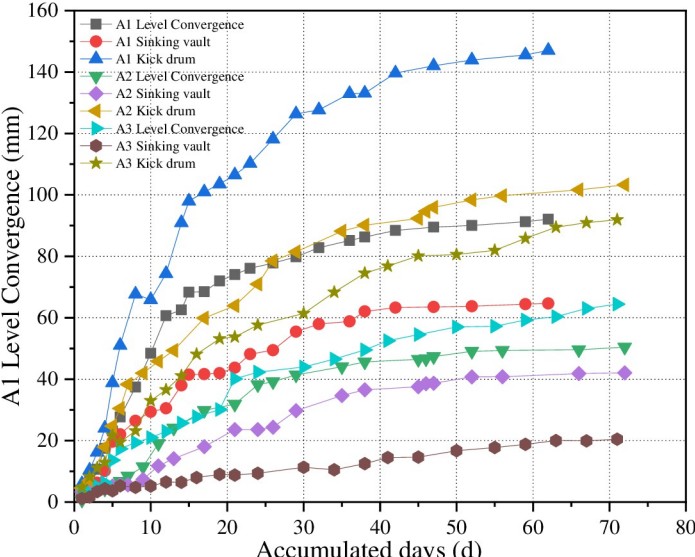

**Fig 9. The surrounding rock deformation-time diagram.**

convergence at two sides is 0.70 mm/d, that of vault subsidence is 0.57 mm/d, and that of floor heave is 1.40 mm/d. In the case of A3 measuring station, the average rate of horizontal convergence at two sides is 0.89 mm/d, that of vault subsidence is 0.30 mm/d, and that of floor heave is 1.26 mm/d.

The field measured data indicate that the reinforced support in the fault-affected stage can effectively reduce the deformation of roadway surrounding rock. To be specific, at the beginning of excavation, the displacement rate of surrounding rock increases rapidly, and then decreases with the increase of distance from the fault; when the excavation lasts for 30 days, the surface displacement rate of surrounding rock begins to decrease; when the excavation continues till 50~60 days, the roadway deformation basically tends to be stable; and at this time the deformation rate is within the normal control range of 0.1~0.5 mm/d.

## 6. Conclusions

(1) By introducing Hoek-Brown strength criterion to analyze the stress distribution in the elastic-plastic zone, this paper reveals that the instability range of roadway surrounding rock is exponentially correlated with the rupture degree of surrounding rock under the action of fault zone. On this basis, the critical instability range of roadway surrounding rock is deduced to be 2.32 m.

(2) Similar experiments show that, compared with ordinary mining, through-the-fault mining causes slippage and dislocation of the fault, the load of the overburden is transferred to both sides of the fault, and the stress near the fault accumulates abnormally. The "three zones" characteristics of the overburden movement basically disappear, the subsidence pattern is changed from "trapezoid" to "inverted triangle", and the influence distance of the advance mining stress on the working face is extended from 20m to 30m.

(3) The engineering application shows that the reinforced support has a good support effect for the roadway under the influence of the fault in the mining process of the working face. At the beginning of excavation, the displacement rate of surrounding rock increases

rapidly, and then decreases with the increase of distance from the fault; when the excavation lasts for 30 days, the surface displacement rate of surrounding rock begins to decrease; when the excavation continues till 50﹍60 days, the roadway deformation basically tends to be stable; and at this time the deformation rate is within the normal control range of 0.1﹍0.5 mm/d.

## Supporting information

**S1 Data. Normal excavation.**
(DOCX)

**S2 Data. Through-the-fault excavation.**
(DOCX)

**S3 Data. The surrounding rock deformation-time diagram.**
(DOCX)

## Author Contributions

**Conceptualization:** Chunlin Zeng, Yuejin Zhou, Leiming Zhang, Donggui Mao.

**Data curation:** Chunlin Zeng, Leiming Zhang, Donggui Mao, Kexin Bai.

**Formal analysis:** Chunlin Zeng, Yuejin Zhou.

**Methodology:** Chunlin Zeng, Yuejin Zhou, Donggui Mao.

**Writing – original draft:** Chunlin Zeng.

**Writing – review & editing:** Chunlin Zeng, Yuejin Zhou.

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
