## [Decision Letter · Decision Letter 0]

22 Sep 2021

PONE-D-21-26499Study on overburden failure law and surrounding rock deformation control technology of mining through fault

PLOS ONE

Dear Dr. ZHOU,

The reviewers have commented on your above paper. They indicated it requires some modifications in order to be acceptable for publication

After careful consideration, we feel that it has merit but does not fully meet PLOS ONE’s publication criteria as it currently stands. Therefore, we invite you to submit a revised version of the manuscript that addresses the points raised during the review process.

If you feel that you can suitably address the reviewers' comments (included below), I invite you to revise and resubmit your manuscript.

Please carefully address the issues raised in the comments. 

We look forward to receiving your revised manuscript.

Kind regards,

Fabio Trippetta, Ph.D.

Academic Editor

PLOS ONE

 “This research was supported by the Fundamental Research Funds for the Central Universities (2019ZDPY18), and project supported by the National Natural Science Foundation of China (51874289).”

4. Thank you for stating the following in the Funding Section of your manuscript:

“This research was supported by the Fundamental Research Funds for the Central Universities (2019ZDPY18), and project supported by the National Natural Science Foundation of China (51874289).”

We note that you have provided information within the Funding Section. Please note that funding information should not appear in other areas of your manuscript. We will only publish funding information present in the Funding Statement section of the online submission form.

“This research was supported by the Fundamental Research Funds for the Central Universities (2019ZDPY18), and project supported by the National Natural Science Foundation of China (51874289).”

7. Please amend the manuscript submission data (via Edit Submission) to include author “Kexin Bai”.

Reviewers' comments:

Reviewer's Responses to Questions

**Comments to the Author**

1. Is the manuscript technically sound, and do the data support the conclusions?

Reviewer #1: Yes

Reviewer #2: Yes

2. Has the statistical analysis been performed appropriately and rigorously? 

Reviewer #1: I Don't Know

Reviewer #2: Yes

3. Have the authors made all data underlying the findings in their manuscript fully available?

Reviewer #1: No

Reviewer #2: Yes

4. Is the manuscript presented in an intelligible fashion and written in standard English?

Reviewer #1: Yes

Reviewer #2: Yes

5. Review Comments to the Author

Reviewer #1: 1.What is the study object of the paper? Working face or roadway?

2. The Abstract section is suggested to re-write according to the study objective.

3. Please clarify the novelty of the MS.

4. In physical model, why is there no roadway？If you study the surrounding rock deformation control technology of roadway, the roadway shoulde be arrangment in physical model.

5. In section 5.2, please check the reliability of result of 2.32m。

6. In Fig.5b, why two peak stress point with the No.7 205cm and the peak value of rear face exceeds that of front value?

7. What is the basis for setting the two support schemes for the roadway with different mining conditions in section 5.2.

Reviewer #2: The manuscript is based on the theoretical analysis and similar material simulation test, the characteristics of roadway deformation, overburden movement and stress evolution in the process of working face fault advance are studied, and the formula for calculating the critical instability range of roadway surrounding rock is derived. The method is applied to engineering practice, which is well verified by field practice and has certain theoretical and practical value. However, there are still some problems in the manuscript, hoping to help improve the quality of the manuscript:

1.It is suggested to put "On this basis, the critical instability range of open-cut surrounding rock is deduced to be 2.32m" in the penultimate sentence of the abstract. Because the critical instability range of open-cut surrounding rock is obtained by inserting the parameters obtained from the simulation of similar materials into the formula. This order makes summaries more organized and readable.

2.The introduction part only lists the research status at home and abroad, but does not get its own understanding and views, and lacks analysis and summary. In addition, the literature tends to be old and should be supplemented with recent research results.

3.Equation (16) can be reduced to the equation with N, since we have set.

4.It is suggested to combine equations (20) and (21) into one set of equations.

5.It is mentioned in the overview of the working face that two faults are actually exposed, but this paper only studies the failure characteristics of surrounding rock over one fault. The two faults interact with each other during exploitation? Please explain.

6.The page 16, line 187, a force equivalent to the overlay load is applied to the upper surface of the model. Please instructions the thickness of the covering layer. What is the magnitude of the equal effect?

7.Please add a plan of the working face in the general section of the working face, indicating the position of the working face in the mining area, the fault position and the position of three measuring stations A1, A2 and A3, etc.

8.Fig.6 and Fig.7 please add the legend part to illustrate the meanings of different lines and marks in the figure.

9.Conclusions should be drawn back to the original text and it is suggested to indicate from which part each conclusion was drawn. For example, numerical simulation experiments show that overburden movement shows obvious "three zones......"

10.The format of 24 references is not correct.

6. PLOS authors have the option to publish the peer review history of their article (what does this mean?). If published, this will include your full peer review and any attached files.

Reviewer #1: No

Reviewer #2: No

---

## [Author Response · Author response to Decision Letter 0]

11 Oct 2021

Dear Editors and Reviewers, 

Thank you very much for your careful review of our paper. The feedback is very valuable. We have revised the manuscript in response to your suggestions and questions. All the modifications made according to reviewer’s comments are highlighted in yellow and using track changes in the manuscript. Our responses are also outlined below following your comments. I hope the revised manuscript is acceptable for publication in our journal. 

Sincerely,

Yuejin Zhou, Ph. D.

Corresponding author for the manuscript

 

Ref: PONE-D-21-26499

Title: Study on overburden failure law and surrounding rock deformation control technology of mining through fault

Journal: PLOS ONE

Answer: Thank you for this suggestion. We have tried our best to revise the manuscript and file according to the requirements of our journal.

Answer: Your help is greatly appreciated. We revised the numbers for the awards. This research was supported by the Fundamental Research Funds for the Central Universities (2019ZDPY18 to JYZ) and the National Natural Science Foundation of China (51874289 to JYZ).

3. Thank you for stating the financial disclosure. Please state what role the funders took in the study.

Answer: Thank you for this suggestion. The funders had no role in study design, data collection and analysis, decision to publish, or preparation of the manuscript.

4. Please remove any funding-related text from the manuscript and let us know how you would like to update your Funding Statement.

Answer: Thank you for this suggestion. We have removed any funding-related text from the manuscript.

5. Upon re-submitting your revised manuscript, please upload your study’s minimal underlying data set as either Supporting Information files or to a stable, public repository and include the relevant URLs, DOIs, or accession numbers within your revised cover letter. 

Answer: Thank you for this suggestion. We have uploaded experimental data in the manuscript and set it as Supporting Information files.

6. Please ensure that you have an ORCID iD and that it is validated in Editorial Manager.

Answer: Thank you for this suggestion. I have set up one ORCID iD, and it is validated in Editorial Manager.

7. Please amend the manuscript submission data (via Edit Submission) to include author “Kexin Bai”

Answer: Thank you for this suggestion. We have tried our best to revised the manuscript submission data accordingly.

Reviewer #1:

1.What is the study object of the paper? Working face or roadway?

Answer: Thanks for your comment. The main research object of this paper is the roadway affected by faults in the mining process of the working face. 

2. The Abstract section is suggested to re-write according to the study objective.

Answer: Thank you for this suggestion. We have rewritten the abstract section according to the study objective in line 11-29 of the revised manuscript. The abstract is rewritten as follows: “In the mining process of working face, the additional stress generated by the fault changes the law of roadway deformation and failure as well as the law of overburden failure. Aiming at the influence of the fault in the mining process of working face, this study introduced the Hoek-Brown strength criterion to analyze the stress distribution in the elastic-plastic zone of the surrounding rock of the roadway. And similar experiments under different engineering backgrounds were used to study the characteristics of overburden movement and stress evolution. This study shows that, compared with ordinary mining, through-the-fault mining causes slippage and dislocation of the fault, the load of the overburden is transferred to both sides of the fault, and the stress near the fault accumulates abnormally. The “three zones” characteristics of the overburden movement disappear, the subsidence pattern is changed from "trapezoid" to "inverted triangle", and the influence distance of the advanced mining stress on the working face is extended from 20m to 30m. The instability range of roadway surrounding rock is exponentially correlated with the rupture degree of the surrounding rock. On this basis, the critical instability range of roadway surrounding rock is deduced to be 2.32m. According to the conclusion, the bolt length and roadway reinforced support length are redesigned. Engineering application shows that the deformation rate of the roadway within 60 days is controlled below 0.1~0.5mm/d, the deformation amount is controlled within 150mm, and the roadway deformation is controlled, which generally meets the requirements of use. The research results provide guidance and reference for similar roadway support.”

3. Please clarify the novelty of the MS.

Answer: Aiming at the impact of the fault on the roadway in the mining process of working face, this paper analyzes the elastoplastic zone of the roadway and the failure law of the overlying rock layer through a combination of theory and similar experiments under different backgrounds. This study shows that 1) The instability range of roadway surrounding rock is exponentially correlated with the rupture degree of surrounding rock. On this basis, the critical instability range of roadway surrounding rock is deduced to be 2.32m. 2) Compared with ordinary mining, through-the-fault mining causes slippage and dislocation of the fault, the load of the overburden is transferred to both sides of the fault, and the stress near the fault accumulates abnormally. The “three zones” characteristics of the overburden movement basically disappear, the subsidence pattern is changed from "trapezoid" to "inverted triangle", and the influence distance of the advance mining stress on the working face is extended from 20m to 30m. 3) According to the conclusion, the bolt length and roadway reinforced support length are redesigned. Engineering application shows that the deformation rate of the roadway within 60 days is controlled below 0.1~0.5mm/d, the deformation amount is controlled within 150mm, and the roadway deformation is controlled, which generally meets the requirements of use. The research results provide guidance and reference for similar roadway support design.

4. In physical model, why is there no roadway？If you study the surrounding rock deformation control technology of roadway, the roadway shoulde be arrangment in physical model.

Answer: Thanks for your comment. In order to study the overburden failure law and the length range under the influence of the fault during the mining process of the working face, similar experiments were carried out by using the plane stress model. Since the roadway is in the same direction as the working face, so no roadway is arranged. We have added the description of similar experiment settings in line 180-183 of the revised manuscript. 

5. In section 5.2, please check the reliability of result of 2.32m.

Answer: Thanks for your comment. We have partially merged the formulas in line 175 of the revised manuscript, Modify “ The equation（21）” to “The equation（20） ”. and the specific calculation results are written in the file of“Responses to reviewers”

6. In Fig.5b, why two peak stress point with the No.7 205cm and the peak value of rear face exceeds that of front value?

Answer: Thanks for your comment. Due to the activation of the fault, another stress peak appears in the vertical stress of the hanging wall of the fault. The hanging wall and foot wall of the fault are dislocated and separated, and the stress on both sides of the fault changes. During the advancement of the working face, the subsidence of the overlying rock layer changes from “trapezoid” to “inverted triangle”. A large amount of load migrates to the coal and rock mass of the hanging wall of the fault, causing the peak stresses at the measuring points of the hanging wall to be higher than that of the foot wall. And we have added relevant explanations in line 317-326 of the revised manuscript.

7. What is the basis for setting the two support schemes for the roadway with different mining conditions in section 5.2.

Answer: Thanks for your comment. According to theoretical calculations and similar experiments, the instability range and influence distance of the roadway under the influence of mining and the fault have been obtained, and the bolt length and roadway reinforced support length are redesigned. We have added relevant explanations in line 367-370 of the revised manuscript.

Reviewer #2:

The manuscript is based on the theoretical analysis and similar material simulation test, the characteristics of roadway deformation, overburden movement and stress evolution in the process of working face fault advance are studied, and the formula for calculating the critical instability range of roadway surrounding rock is derived. The method is applied to engineering practice, which is well verified by field practice and has certain theoretical and practical value. However, there are still some problems in the manuscript, hoping to help improve the quality of the manuscript:

1.It is suggested to put "On this basis, the critical instability range of open-cut surrounding rock is deduced to be 2.32m" in the penultimate sentence of the abstract. Because the critical instability range of open-cut surrounding rock is obtained by inserting the parameters obtained from the simulation of similar materials into the formula. This order makes summaries more organized and readable.

Answer: Thank you for this suggestion. We have rewritten the abstract section and put "On this basis, the critical instability range of open-cut surrounding rock is deduced to be 2.32m" behind the conclusions of similar experiments, as showed in line 23-24 of the revised manuscript.

2.The introduction part only lists the research status at home and abroad, but does not get its own understanding and views, and lacks analysis and summary. In addition, the literature tends to be old and should be supplemented with recent research results.

Answer: Thank you for this suggestion. We have updated references in line 64 and 74, and added “These studies are based on the complex rock and soil environment under the influence of faults on the roadway failure. Due to the complex and changeable engineering geological conditions, the complex rock and soil environment and the impact of mining on the surrounding rock under the influence of faults are the main reasons for roadway failure. In order to ensure the stability of the surrounding rock of the roadway under the superimposed influence of faults and mining, it is necessary to study its failure mechanism, and then propose a reasonable support design.” in line 70-76 of the introduction section.

3.Equation (16) can be reduced to the equation with N, since we have set.

Answer: Thanks for your comment. We have simplified equation (16) to the equation with N in line 162 of the revised manuscript. 

4.It is suggested to combine equations (20) and (21) into one set of equations.

Answer: Thank you for this suggestion. We have combined equations (20) and (21) into one set of equations in line 173-175 of the revised manuscript. 

5.It is mentioned in the overview of the working face that two faults are actually exposed, but this paper only studies the failure characteristics of surrounding rock over one fault. The two faults interact with each other during exploitation? Please explain.

Answer: Thanks for your comment. The distance between the two faults is 387.6m. Because they are far apart, they will not affect each other during the mining process of the working face. We have added description of the mining face in line 351-353 of the revised manuscript, and added an overview map and as shown in Fig 6.

6.The page 16, line 187, a force equivalent to the overlay load is applied to the upper surface of the model. Please instructions the thickness of the covering layer. What is the magnitude of the equal effect?

Answer: Thanks for your comment. We have replaced “A counterweight equivalent to the overburden load is applied to the upper surface of the model.” with “According to the thickness of the covering layer is 428.11m and the stress similarity ratio is 91.88, a stress of 0.09 Mpa is applied to the upper surface of the model to compensate for the influence of the overburden load on the model.”, as showed in line 196-198 of the revised manuscript.

7.Please add a plan of the working face in the general section of the working face, indicating the position of the working face in the mining area, the fault position and the position of three measuring stations A1, A2 and A3, etc.

Answer: Thanks for your comment. We have added a plan of the working face, indicating the position of the working face in the mining area, the fault position and the position of three measuring stations A1, A2 and A3, as shown in Fig 6 of the revised manuscript. 

8.Fig.6 and Fig.7 please add the legend part to illustrate the meanings of different lines and marks in the figure.

Answer: Thanks for your comment. We have added the legend part to illustrate the meanings of different lines and marks as showed in Fig 7 and Fig 8 of the revised manuscript.

9.Conclusions should be drawn back to the original text and it is suggested to indicate from which part each conclusion was drawn. For example, numerical simulation experiments show that overburden movement shows obvious "three zones......"

Answer: Thanks for your comment. We have revised the conclution and indicated from which part each conclusion was drawn, as showed in the conclution part of the revised manuscript.

Modify “Under normal mining conditions, overburden movement displays obvious three-zone characteristics and presents trapezoidal subsidence, and the influence distance of mining stress in advance of working face is about 20 m. Under through-the-fault mining conditions, affected by the inherent characteristics of fault, the stress increases abnormally near the fault. Fault slip and dislocation cause overburden load to transfer to both sides of the fault. Abnormal stress accumulation occurs, and the “three zone” characteristics of overburden movement basically disappear. The hanging wall subsides in the form of inverted triangle, and the influence distance of the advance mining stress on the working face is about 30 m from the fault.” to “Similar experiments show that, compared with ordinary mining, through-the-fault mining causes slippage and dislocation of the fault, the load of the overburden is transferred to both sides of the fault, and the stress near the fault accumulates abnormally. The “three zones” characteristics of the overburden movement basically disappear, the subsidence pattern is changed from "trapezoid" to "inverted triangle", and the influence distance of the advance mining stress on the working face is extended from 20m to 30m.”

Modify “In terms of the roadway under the influence of fault, the reinforced support has a better supporting effect, which verifies the accuracy of the conclusion.” to “The engineering application shows that the reinforced support has a good support effect for the roadway under the influence of the fault in the mining process of the working face.”

10.The format of 24 references is not correct.

Answer: Thanks for your comment. We have revised the manuscript to the requirements of our journal as showed in the reference part of the revised manuscript.

---

## [Decision Letter · Decision Letter 1]

16 Nov 2021

PONE-D-21-26499R1Study on overburden failure law and surrounding rock deformation control technology of mining through faultPLOS ONE

Dear Dr. ZHOU,

Thank you for submitting your manuscript to PLOS ONE. After careful consideration, we feel that it has merit but does not fully meet PLOS ONE’s publication criteria as it currently stands. Therefore, we invite you to submit a revised version of the manuscript that addresses the points raised during the review process.

In particular, reviewers pointed out that the paper's novelty should be clarified. Moreover they also highlight the need of clarifying some technical steps. I encourage the authors in carefully following the reviewers suggestions, since I think that they will greatly improve the paper.

We look forward to receiving your revised manuscript.

Kind regards,

Fabio Trippetta, Ph.D.

Academic Editor

PLOS ONE

Journal Requirements:

Reviewers' comments:

Reviewer's Responses to Questions

**Comments to the Author**

1. If the authors have adequately addressed your comments raised in a previous round of review and you feel that this manuscript is now acceptable for publication, you may indicate that here to bypass the “Comments to the Author” section, enter your conflict of interest statement in the “Confidential to Editor” section, and submit your "Accept" recommendation.

Reviewer #1: (No Response)

Reviewer #2: All comments have been addressed

2. Is the manuscript technically sound, and do the data support the conclusions?

Reviewer #1: Yes

Reviewer #2: Yes

3. Has the statistical analysis been performed appropriately and rigorously? 

Reviewer #1: Yes

Reviewer #2: Yes

4. Have the authors made all data underlying the findings in their manuscript fully available?

Reviewer #1: Yes

Reviewer #2: Yes

5. Is the manuscript presented in an intelligible fashion and written in standard English?

Reviewer #1: Yes

Reviewer #2: Yes

6. Review Comments to the Author

Reviewer #1: 1. Please continuously clarify the novelty of the MS because the MS does not well show the novelty.

2. please check the reliability of result of 2.32m. In eq.20, the values of s shows 0.1 and 0.01, please check it.

3. It is suggested to remove the author marks of equally contuibution.

Reviewer #2: All the revisions are addressed.

What's the groundwater impact the fault movement? does it influnce you results?

7. PLOS authors have the option to publish the peer review history of their article (what does this mean?). If published, this will include your full peer review and any attached files.

Reviewer #1: No

Reviewer #2: No

---

## [Author Response · Author response to Decision Letter 1]

19 Nov 2021

Dear Editors and Reviewers, 

Thank you very much for your careful review of our paper. Those comments are all valuable and very helpful for revising and improving our paper, as well as the important guiding significance to our researches. We have revised the manuscript in response to your suggestions and questions. All the modifications made according to reviewer’s comments are highlighted in yellow and using track changes in the manuscript. Our responses are also outlined below following your comments. I hope the revised manuscript is acceptable for publication in our journal. 

Sincerely,

Yuejin Zhou, Ph. D.

Corresponding author for the manuscript

 

Ref: PONE-D-21-26499R1

Title: Study on overburden failure law and surrounding rock deformation control technology of mining through fault

Journal: PLOS ONE

Reviewer #1:

1. Please continuously clarify the novelty of the MS because the MS does not well show the novelty.

Answer: Thanks for your comment. Aiming at the influence of mining face excavation on the roadway through fault, the geological strength index (GSI) was introduced to obtain the basis for judging the instability of the surrounding rock of the roadway through fault, and combined with two different similar tests to analyze the scope of roadway instability and the failure law of overlying rock masses. According to the conclusions obtained, the roadway support scheme is designed. This study shows that: 1) Through the introduction of GSI, it is found that the instability range of roadway surrounding rock is exponentially correlated with the rupture degree of surrounding rock. On this basis, the critical instability range of roadway surrounding rock is deduced to be 2.32m. 2) Compared with ordinary mining, through-the-fault mining causes slippage and dislocation of the fault, the load of the overburden is transferred to both sides of the fault, and the stress near the fault accumulates abnormally. The “three zones” characteristics of the overburden movement basically disappear, the subsidence pattern is changed from "trapezoid" to "inverted triangle", and the influence distance of the advance mining stress on the working face is extended from 20m to 30m. 3) According to the conclusion, the bolt length and roadway reinforced support length are redesigned. Engineering application shows that the deformation rate of the roadway within 60 days is controlled below 0.1~0.5mm/d, the deformation amount is controlled within 150mm, and the roadway deformation is controlled, which generally meets the requirements of use. The research results provide guidance and reference for similar roadway support design. 

2. please check the reliability of result of 2.32m. In eq.20, the values of s shows 0.1 and 0.01, please check it.

Answer: Thank you very much for your careful review of our paper. The value of s is 0.01, we are very sorry that we missed the number when editing the formula. The overall calculation result is correct. and the specific calculation results are written in the file of“Responses to reviewers” .

3. It is suggested to remove the author marks of equally contuibution.

Answer: Thank you for this suggestion. We have removed the author marks of equally contuibution.

Reviewer #2:

1. What's the groundwater impact the fault movement? does it influnce your results? 

Answer: Thanks for your comment. This paper studies the impact of mining face excavation on the roadway through fault. The activation of faults is mainly caused by mining face excavation, and the influence of groundwater in this process is small and does not influnce the results.

---

## [Editor Report · Decision Letter 2]

2 Dec 2021

PONE-D-21-26499R2Study on overburden failure law and surrounding rock deformation control technology of mining through faultPLOS ONE

Dear Dr. ZHOU,

Thank you for submitting your manuscript to PLOS ONE. After careful consideration, we feel that it has merit but does not fully meet PLOS ONE’s publication criteria as it currently stands. Therefore, we invite you to submit a revised version of the manuscript that addresses the points raised during the review process.

I noticed that the author answered to the reviewer's comment only in the "Responses to reviewers" file. However the "Revised Manuscript with Track Changes" show no changes with respect to the previous version. I would like to highlight that if a reviewer ask a question this means that that message is not clear along the paper and, thus, the answer should be added to the paper in order to clarify that point.  Consequently, I suggest the authors to clarify the questions raised by the reviewers directly in the paper. For clarity I report again the previous main reviewers points: 1. Please continuously clarify the novelty of the MS because the MS does not well show the

novelty.2. What's the groundwater impact the fault movement? does it influnce your results?

We look forward to receiving your revised manuscript.

Kind regards,

Fabio Trippetta, Ph.D.

Academic Editor

PLOS ONE
---

## [Author Response · Author response to Decision Letter 2]

7 Dec 2021

Dear Editors and Reviewers, 

Thank you very much for your reminder, and thank you very much for your careful review of our paper. Those comments are all valuable and very helpful for revising and improving our paper, as well as the important guiding significance to our researches. We have revised the manuscript in response to your suggestions and questions. All the modifications made according to the reviewer’s comments are highlighted in yellow and using track changes in the manuscript. Our responses are also outlined below following your comments. I hope the revised manuscript is acceptable for publication in our journal. Finally, thank you again for your reminders and comments.

Sincerely,

Yuejin Zhou, Ph. D.

Corresponding author for the manuscript

 

Ref: PONE-D-21-26499

Title: Study on overburden failure law and surrounding rock deformation control technology of mining through fault

Journal: PLOS ONE

Reviewer #1:

1. Please continuously clarify the novelty of the MS because the MS does not well show the novelty.

Answer: Thanks for your comment. Your comment is very helpful for revising and improving our paper. We have revised the abstract section to better express the novelty of the MS in lines 11-15 and line 22 of the revised manuscript. The revised summary section is as follows：

In the mining process of working face, the additional stress generated by the fault changes the law of roadway deformation and failure as well as the law of overburden failure. Aiming at the influence of the fault in the mining process of working face, this study introduced the geological strength index (GSI) to analyze the stress distribution in the elastic-plastic zone of the surrounding rock of the roadway. And similar experiments under different engineering backgrounds were combined to study the characteristics of overburden movement and stress evolution. Based on the conclusions obtained, the roadway support scheme was designed. This study shows that, compared with ordinary mining, through-the-fault mining causes slippage and dislocation of the fault, the load of the overburden is transferred to both sides of the fault, and the stress near the fault accumulates abnormally. The “three zones” characteristics of the overburden movement disappear, the subsidence pattern is changed from "trapezoid" to "inverted triangle", and the influence distance of the advanced mining stress on the working face is extended from 20m to 30m. The instability range of roadway surrounding rock is exponentially correlated with the rupture degree of the surrounding rock. Through the introduction of GSI, the critical instability range of roadway surrounding rock is deduced to be 2.32m. According to the conclusion, the bolt length and roadway reinforced support length are redesigned. Engineering application shows that the deformation rate of the roadway within 60 days is controlled below 0.1~0.5mm/d, the deformation amount is controlled within 150mm, and the roadway deformation is controlled, which generally meets the requirements of use. The research results provide guidance and reference for similar roadway support.

2. please check the reliability of result of 2.32m. In eq.20, the values of s shows 0.1 and 0.01, please check it.

Answer: Thank you very much for your careful review of our paper. The value of s is 0.01, we are very sorry that we missed the number when editing the formula. The overall calculation result is correct. And the specific calculation results are written in the file of“Responses to reviewers” . 

3. It is suggested to remove the author marks of equally contuibution.

Answer: Thank you for this suggestion. After our discussion, we have removed the author marks of equally contuibution in line 3 of the revised manuscript.

Reviewer #2:

1. What's the groundwater impact the fault movement? does it influnce your results? 

Answer: Thanks for your comment. Your comment is very valuable for revising our paper. Generally, when a fault zone connects to aquifers, groundwater will spread to the fault. The pressure of pore fluid in the fault zone will increase, and the friction strength of the fault will decrease, which will accelerate fault instability. For this study, since the fault has a small drop and does not connect to the aquifer, the excavation of the working face under the influence of the fault is the main reason for roadway instability, so the groundwater is not considered. We have added relevant explanations in lines 188-190 of the revised manuscript. The distribution law of fault stress under the influence of groundwater, involves water-sediment seepage, the hydration reaction of fault, and the impact on roadway instability. We are researching these aspects, and we have achieved some results so far. We hope to publish relevant research results on PLOS ONE in the future.

---

## [Editor Report · Decision Letter 3]

20 Dec 2021

Study on overburden failure law and surrounding rock deformation control technology of mining through fault

PONE-D-21-26499R3

Dear Dr. ZHOU,

We’re pleased to inform you that your manuscript has been judged scientifically suitable for publication and will be formally accepted for publication once it meets all outstanding technical requirements.

Kind regards,

Fabio Trippetta, Ph.D.

Academic Editor

PLOS ONE
---

## [Editor Report · Acceptance letter]

10 Jan 2022

PONE-D-21-26499R3 

Study on overburden failure law and surrounding rock deformation control technology of mining through fault 

Dear Dr. Zhou:

I'm pleased to inform you that your manuscript has been deemed suitable for publication in PLOS ONE. Congratulations! Your manuscript is now with our production department. 

Kind regards, 

on behalf of

Prof. Fabio Trippetta 

Academic Editor

PLOS ONE